

# Recurrent attention U-Net for segmentation and quantification of breast arterial calcifications on synthesized 2D mammograms

Manar AlJabri[1,2], Manal Alghamdi[1], Fernando Collado-Mesa[3] and Mohamed Abdel-Mottaleb[4]

[1] Department of Computer Science and Artificial Intelligence, Umm Al-Qura University, Makkah, Makkah, Saudi Arabia
[2] King Abdul Aziz University, Jeddah, Makkah, Saudi Arabia
[3] Department of Radiology, Miller School of Medicine, University of Miami, Miami, Florida, United States
[4] Department of Electrical and Computer Engineering, University of Miami, Miami, Florida, United States

Corresponding author
Manal Alghamdi,
maalghamdi@uqu.edu.sa

## ABSTRACT

Breast arterial calcifications (BAC) are a type of calcification commonly observed on mammograms and are generally considered benign and not associated with breast cancer. However, there is accumulating observational evidence of an association between BAC and cardiovascular disease, the leading cause of death in women. We present a deep learning method that could assist radiologists in detecting and quantifying BAC in synthesized 2D mammograms. We present a recurrent attention U-Net model consisting of encoder and decoder modules that include multiple blocks that each use a recurrent mechanism, a recurrent mechanism, and an attention module between them. The model also includes a skip connection between the encoder and the decoder, similar to a U-shaped network. The attention module was used to enhance the capture of long-range dependencies and enable the network to effectively classify BAC from the background, whereas the recurrent blocks ensured better feature representation. The model was evaluated using a dataset containing 2,000 synthesized 2D mammogram images. We obtained 99.8861% overall accuracy, 69.6107% sensitivity, 66.5758% F-1 score, and 59.5498% Jaccard coefficient, respectively. The presented model achieved promising performance compared with related models.

## INTRODUCTION

Screening mammography is a low-dose X-ray examination of a woman's breasts performed to detect breast cancer. However, the main drawback of standard full-field digital mammography is that overlaying dense fibroglandular tissue within the breast can reduce the visibility of masses and lead to ineffective screening results for breast cancer. Therefore, digital breast tomosynthesis (DBT) is becoming the new standard for x-ray imaging of the breast for breast cancer screening. Multiple studies have shown that when

DBT is combined with conventional digital mammography (DM), both sensitivity and specificity are improved, with a reduction in false positive recalls and an increase in breast cancer detection (*Friedewald et al., 2014*; *Durand et al., 2015*; *Conant et al., 2016*). However, because the combination of DM and DBT incurs a higher radiation dose (still within the acceptable range) than DM alone, the Food and Drug Administration has approved using a reconstructed "synthetic" 2D-like image instead of the DM dose portion, which reduces the overall radiation dose by up to 45% (*Skaane et al., 2013*). Given the increase in the clinical use of DBT, there is a growing research interest in the use of deep learning (DL) to develop detection and segmentation methods for DBT (*Samala et al., 2016*; *Yousefi, Krzyżak & Suen, 2018*). Calcifications of the breast are small calcium deposits that occur in a woman's breast tissue and are visible on mammography (*Aljabri & AlGhamdi, 2022*). Breast arterial calcifications (BAC), are among the different typically benign calcifications noted on mammograms. BAC are not associated with breast cancer; as such, they are not routinely reported by radiologists. On mammographic images, BAC appear as densities in a linear distribution, some straight and some winding, and commonly occur as parallel tracks associated with blood vessels. BAC can range in density, from faint low density to very high density, and may be present from only a few to many in numbers. Several studies have provided evidence that BAC constitutes a risk marker for coronary artery disease, and appear to be associated with an increased risk of cardiovascular disease (CVD) events (*Iribarren et al., 2004*; *Ferreira, Szejnfeld & Faintuch, 2007*; *Maas et al., 2007*; *Iribarren & Molloi, 2013*; *Chadashvili et al., 2016*; *Yoon et al., 2019*; *Newallo et al., 2015*; *Margolies et al., 2016*). BAC identified by mammography in women is a new method for diagnosing CVD and coronary artery disease, adding incremental prognostic value beyond the existing CVD risk classification schemes without further radiation doses or costs.

Automatic segmentation methods enhance clinical workflows by accurately handling tedious tasks, such as outlining large, clear lesions, allowing physicians to focus on diagnosis and patient care. There have been efforts for the automatic segmentation of BAC. *Ge et al. (2008)* developed a method using an image filtering technique to detect BAC, followed by k-segment clustering to discover a group of potential candidates of BAC lines, and a classifier to reduce false positives. The remaining line segments were connected to generate BAC. Their method failed to detect highly winding BAC, which are common. *Cheng et al. (2009)*, *Cheng, Chen & Shen (2012)* and *Cheng et al. (2012)* developed a system that included calcification and vessel cues. The vessel cues reduced the search from background breast tissue to solely tubular structures, making it easier to find line segments corresponding to probable BAC. The system then connects the segments to extract BAC from mammograms. This method's main weakness is that it can not handle diverse breast compositions with different fat and fibroglandular tissue distributions. *Mordang et al. (2016)* developed a multi-stage method to detect BAC. In the first stage, a cascade classifier detects calcifications. Then the calcifications are segmented by connected elements and classified as possible BAC regions or not. The system then groups the mammograms with potential BAC in the second stage. To examine the detection of BAC, the system extracts a collection of features that have been designed manually, such as topology, shape, and

texture, from the regions. In the final stage, the system removes the segmented BAC from mammograms and then uses a computer-aided diagnosis (CAD) system to diagnose breast cancer. They indicate that a CAD system detects 14% more cancer lesions after removing BAC from mammograms. In spite of all the progress made, BAC segmentation in mammograms remains a challenging task because of BAC's characteristics, such as their narrow structure that runs, along with vessels and their fragmented and varied length and width.

DL has performed remarkably well in medical image segmentation. Convolutional neural networks (CNN) have been applied to medical image segmentation in recent years. They have succeeded in the medical field and have aided in auxiliary diagnoses, such as segmenting vessels from fundus images (*Yan, Yang & Cheng, 2018*), brain tumor segmentation from MRI (*Chen et al., 2020*), and diagnosis of lung infections (*Ouyang et al., 2020*). These models have outstanding feature extraction and feature expression capabilities. In addition, they do not require manual feature extraction or excessive image pre-processing.

Fully convolutional networks (FCN) (*Ronneberger, Fischer & Brox, 2015*) and U-Net (*Long, Shelhamer & Darrell, 2015*) are encoder-decoder architectures used in image segmentation. U-Net was proposed after FCN. Both network architectures are relatively similar, except that the U-Net uses skip connections to connect the encoder and decoder, which improves the U-Net model's performance to obtain more feature information. U-Net-based networks are commonly used in different medical image segmentation techniques, such as optic disc and cup segmentation (*Fu et al., 2018*), brain tumor segmentation (*Chen et al., 2018*), and liver and tumor segmentation from CT images (*Jin et al., 2020*). Several authors have proposed various mechanisms to improve the performance of U-Net, such as the residual mechanism and the attention mechanism. Residual convolutions enhance feature utilization, leading to enhanced performance of the network. Furthermore, it adds depth to the network while ensuring better performance. Researchers have introduced it into the field of medical image segmentation by integrating residual blocks and U-Net architecture for skin image segmentation (*Alom et al., 2018*) and white matter hyperintensity segmentation from MRI (*Jin et al., 2018*). Using the attention mechanism in medical image segmentation has become a hot research topic (*Liu et al., 2021*) and has been used in pancreas segmentation (*Oktay et al., 2018*) and brain structure segmentation (*Li, Zhygallo & Menze, 2018*).

In spite of the advancements in DL networks for medical image segmentation, BAC segmentation remains a formidable challenge due to the intricate nature of mammogram images and the lack of publicly available datasets. Early attempts, such as the CNN architecture proposed by *Wang et al. (2017)*, for BAC segmentation with a patch-based approach, faced limitations in capturing the global context of the images. Subsequent studies have explored various DL models, including YOLO, U-Net, and DeepLabv3+, to improve segmentation accuracy (*Wang, Khan & Highnam, 2019*). Innovations like the DU-Net model introduced by *AlGhamdi, Abdel-Mottaleb & Collado-Mesa (2020)*, which integrates dense blocks into the U-Net architecture, and the use of dilated convolutional layers in the Simple Context U-Net by *Guo et al. (2021)*, have pushed the boundaries

further by enhancing the model's ability to aggregate contextual information. These efforts highlight the ongoing quest to refine segmentation techniques, underscoring the complexity of BAC segmentation and the critical need for more comprehensive solutions.

## Contributions

In this article, we present the results of a novel recurrent attention U-Net to extract BAC structures from mammographic images. Our work was inspired by the success of using attention mechanisms to segment curvilinear structures from different imaging modalities (*Fu et al., 2019*; *Mou et al., 2021*). Moreover, the success of combining a recurrent neural network within the U-Net architecture for vessel segmentation by *Alom et al. (2018)*. The contributions of this work are as follows:

1) We developed a new model for BAC segmentation and quantification, which first generates BAC masks used later to calculate five metrics that indicate the severity level of BAC in the segmentation mask.
2) The new model extended U-Net using recurrent blocks and an attention module to ensure better feature representation and improve the abilities of the U-Net to effectively capture long-range dependencies.
3) We published a new dataset with the ground-truth of BAC, which was annotated by radiologists. This dataset with the annotations will be made available to foster future research in this area.

The article is structured as follows: "Introduction" introduces the method. "Related Work" reviews related work, highlighting significant contributions and identifying gaps in the current literature on DL networks for BAC segmentation. The DL networks for our BAC segmentation model are presented in "Approach". The datasets, quantification metrics, and experimental setup are presented in "Experiments". The experimental results are presented in "Results". The results are discussed in "Discussion". The conclusions are in "Conclusions", and the promising future works are in "Promising Future Works".

## RELATED WORK

DL has performed remarkably well in medical image segmentation. CNN have been used in medical image segmentation in recent years. It has succeeded in the medical field and aided in auxiliary diagnoses. However, despite DL networks excellent results in different medical image segmentation tasks, BAC segmentation remains a challenging task. *Wang et al. (2017)* popularized the use of CNN architecture to build an automated system for BAC segmentation to use it as a sign of coronary artery disease. They developed a model that consists of twelve CNN layers for a binary classification task. They applied a pixel-wise, patch-based procedure for BAC segmentation and used that model to differentiate BAC from non-BAC pixels in mammograms. Therefore, they used an image patch around it to classify the central pixel for any pixel present. In other words, the repetition of the whole image's pixels results in image segmentation. The FROC analysis showed that the model achieved a level of detection similar to that of human radiology experts. The calcium mass quantification analysis showed that the presumed calcium mass is almost equal to the

ground truth. For a linear regression between them, the coefficient of determination (R2) is equal to 0.9624. *Wang, Khan & Highnam (2019)* evaluated the performance of several already known DL models, *i.e.*, YOLO, U-Net, and *DeepLabv*3+, on segmenting BACs in digital mammography. *AlGhamdi, Abdel-Mottaleb & Collado-Mesa (2020)* presented the DU-Net model to segment BAC automatically. They improved the U-Net model by extending it with dense blocks. The DU-Net contains both long-skip connections as the U-Net and short-skip connections as the DenseNet. They prepared a dataset for the BAC segmentation task from a publicly available dataset. Recently, *Guo et al. (2021)* proposed a segmentation method called simple context U-Net (SCU-Net). They used patches to train the model and obtain the final whole image results by joining patchwise outputs. They used five calcifications quantitative metrics to evaluate the results. Instead of using just standard convolution, they used dilated convolutional layers to aggregate multi-scale contextual information. Dilated convolutional layers are a type of convolution that expands the kernel by including gaps between the kernel elements.

The methodology for segmentation in medical images is highly diverse and includes the development of novel CNN-based segmentation structures. U-net networks have played a crucial role in medical image segmentation *via* DL. The combination of U-shaped networks with dense blocks and other mechanisms has led to a variety of deep network architectures, improving medical image segmentation performance significantly. In recent years, end-to-end CNN networks with U-shaped architectures have emerged as the primary method for medical image segmentation. Increasing the depth of network architectures has been shown to enhance network performance. A key factor is the incorporation of doctors' prior understanding of medical imaging, allowing developers to use this knowledge to enhance the performance of segmentation networks by creating accurate ground-truth datasets. Moreover, improving image quality through pre-processing methods can further increase the performance of segmentation networks. Custom architectures developed specifically for segmentation have yielded promising results, often rivaling or even outperforming U-net-based outcomes.

## APPROACH

### Recurrent attention U-Net

This article identifies the BAC segmentation problem as a binary semantic segmentation task, wherein each pixel is categorized into one of two classes. The segmentation task for BAC involved the classification of two different classes: BAC pixels and non-BAC pixels. To address this, we propose an end-to-end approach that is illustrated in Fig. 1. This approach systematically encompasses the entire process, from raw data preparation to the final quantification of BAC. Based on the U-Net (*Ronneberger, Fischer & Brox, 2015*), R2U-Net (*Alom et al., 2018*), and attention module (*Fu et al., 2019*; *Mou et al., 2021*), we propose a recurrent attention U-Net (RAU-Net) for BAC segmentation. The encoder and decoder modules included multiple blocks, each employing a recurrent residual block. An attention module was added between the encoder and the decoder. In order to address the issue of information loss resulting from the use of max-pooling operations, we used a skip connection between every block of the encoder and decoder, similar to the U-Net network.

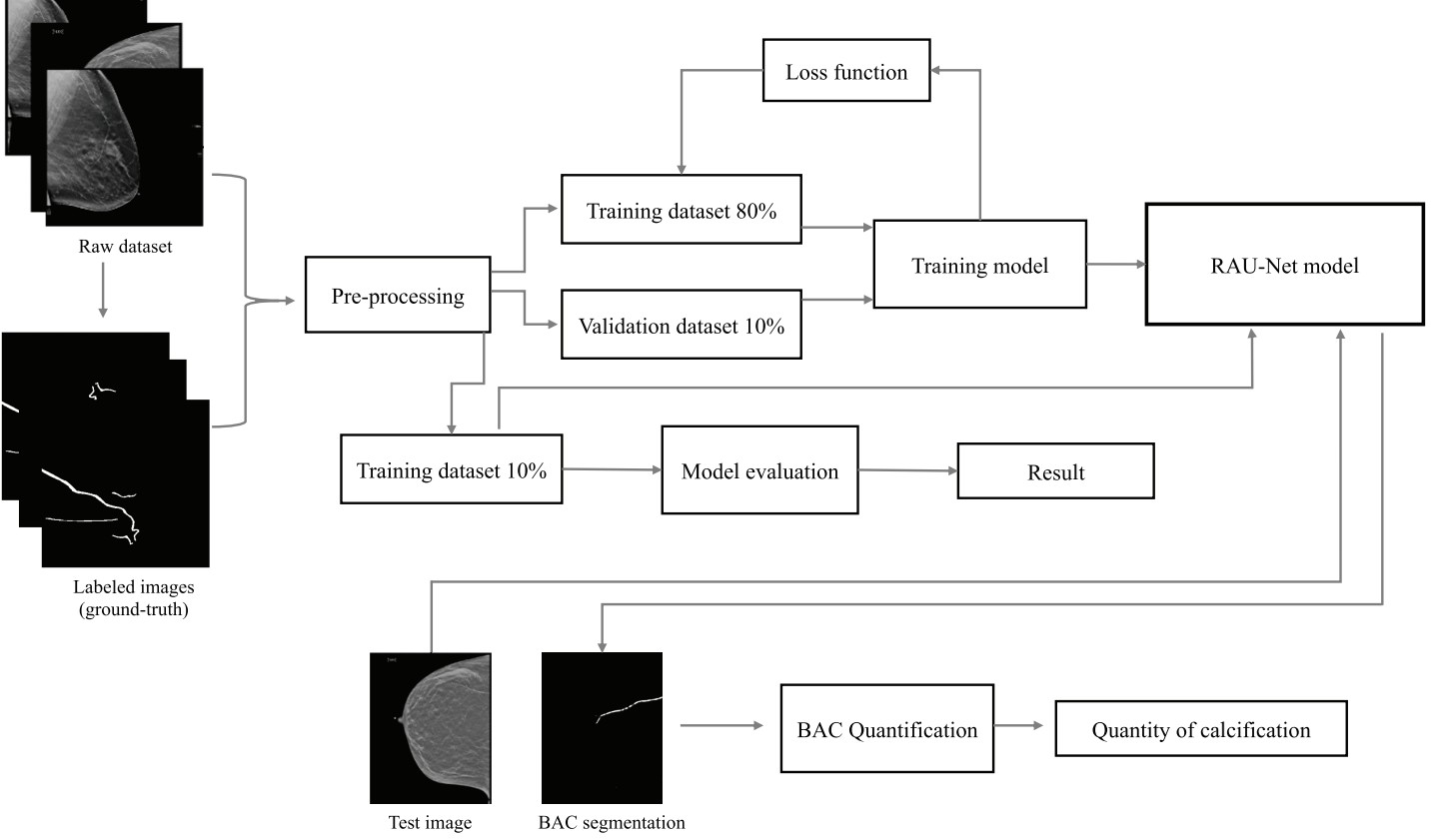

**Figure 1 Overview of the end-to-end proposed approach for BAC segmentation.** This diagram illustrates the workflow, starting from the raw dataset to the final BAC quantification. Mammogram images and binary mask images from our dataset.

To obtain the final segmentation map, we applied a $1 \times 1$ convolutional layer to the decoder output at the end of the model. In the case of using a different loss function than BCEWithLogitsLoss, we added the sigmoid layer to the output. A detailed structure of our network is shown in Fig. 2 and Table 1.

### U-Net

The U-Net encoder contracting path was comprised of multiple convolution operations, and downsampling layers were utilized to reduce the image size and capture context information (*Aljabri & AlGhamdi, 2022*). The expanding path of the decoder was created using convolution operations, and upsampling layers were used to precisely locate pixels and restore images. The architecture of U-Net included convolution, downsampling, upsampling, and concatenation operations. To integrate more contextual information, concatenation operations were used at the same depth between the contracting path and the expanding path.

### Recurrent block

The operations of the recurrent convolutional layers were executed in the manner defined by *Alom et al. (2018)*. Suppose the *x* input sample in the *l*'th layer and the center pixel of a

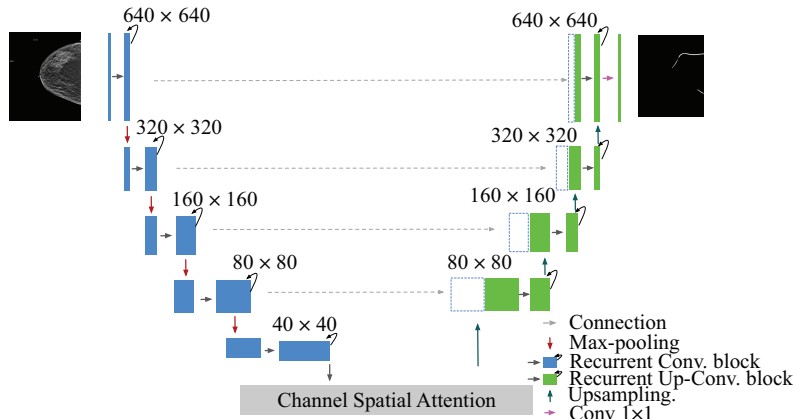

**Figure 2 Architecture of the proposed model RAU-Net with encoder and decoder units using recurrent blocks and an attention module, which is built based on a U-shaped architecture.** Mammogram image and binary mask image from our dataset.

**Table 1 Recurrent attention U-Net architecture, note each convolutional in the recurrent convolutional block in the table performs BatchNormalization + ReLU activation + convolutional.**

| Layers | Output shape |
|---|---|
| Input | $640 \times 640$ |
| Recurrent Conv. block | $640 \times 640$ |
| Max-pooling | $320 \times 320$ |
| Recurrent Conv. block | $320 \times 320$ |
| Max-pooling | $160 \times 160$ |
| Recurrent Conv. block | $160 \times 160$ |
| Max-pooling | $80 \times 80$ |
| Recurrent Conv. block | $80 \times 80$ |
| Max-pooling | $40 \times 40$ |
| Recurrent Conv. block | $40 \times 40$ |
| Spatial attention block | $40 \times 40$ |
| Channel attention block | $40 \times 40$ |
| Affinity attention | $40 \times 40$ |
| Upsampling | $80 \times 80$ |
| Recurrent Up-Conv. block | $80 \times 80$ |
| Upsampling | $160 \times 160$ |
| Recurrent Up-Conv. block | $160 \times 160$ |
| Upsampling | $320 \times 320$ |
| Recurrent Up-Conv. block | $320 \times 320$ |
| Upsampling | $640 \times 640$ |
| Recurrent Up-Conv. block | $640 \times 640$ |
| Conv. $1 \times 1$ | $640 \times 640$ |

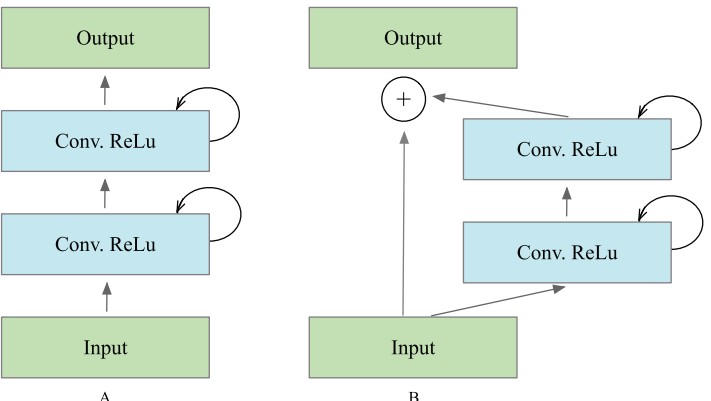

**Figure 3 Different types of recurrent convolutional units (RCUs), which include (A) the recurrent convolutional block and (B) the recurrent residual convolutional unit.**

patch placed at $(i, j)$ in input on sample the $k$'th feature map in the layers, the output of the network $O^l_{ijk}(t)$ is at the time step $t$. The output of the recurrent block is expressed as Eq. (1). $x_l^{f(i,j)}(t)$ and $x_l^{r(i,j)}(t-1)$ are the inputs for the standard convolutional layers and the $l$'th recurrent convolutional layer. The $(W^f_k)$ and $(W^r_k)$ values are the weights for the standard convolutional layer and the recurrent convolutional layer of the $k$'th feature map, and the $b_k$ is the bias. The ReLU activation function $f$ received the outputs of the recurrent convolutional layer and are expressed as in Eq. (2). $\mathcal{F}(x_l, w_l)$ defines the outputs from $l$'th recurrent layer, which was used for down-sampling and up-sampling layers in the convolutional encoding and decoding model. In the case of the recurrent residual block, the output is passed to the residual of the unit, as illustrated in Fig. 3. The block's output can be computed as in Eq. (1). $x_l$ is the block's input samples. The $x_{l+1}$ in the R2U-Net model's encoder and decoder units represent the input for the immediately following sub-sampling or up-sampling layers, and it can be calculated as Eq. (3).

$$O^l_{ijk}(t) = \left(w^f_k\right)^T \times x_l^{f(i,j)}(t) + \left(w^r_k\right)^T \times x_l^{r(i,j)}(t-1) + b_k \tag{1}$$

$$\mathcal{F}(x_l, w_l) = f\left[O^l_{ijk}(t)\right] = \max\left[0, O^l_{ijk}(t)\right] \tag{2}$$

$$x_{l+1} = x_l + \mathcal{F}(x_l, w_l). \tag{3}$$

Collecting features across various time steps enhances the quality and robustness of feature representation. This helps extract very low-level features crucial for segmentation tasks, such as the tubular structure of the vessels.

### Attention block

Expressive features that are attentive to channel and spatial information were generated by combining a channel attention block and a spatial attention block. The spatial attention block uses a selective aggregation mechanism to combine features at each spatial location by giving weights to the features across all spatial locations. This lets the model accurately capture how the data and features are related over long distances, even when they are far

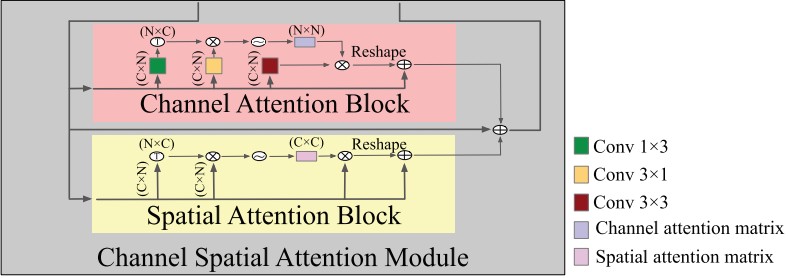

**Figure 4 A detailed structure of the attention module.** Note that each convolutional followed BatchNormalization + ReLU.               

apart. In contrast, the channel attention block ensures that the entire space is used for representing and normalizing; hence, enhancing the incorporation of contrasting features across separate channels enhances the model's ability to discriminate. Figure 4 shows the detailed structure of an attention module where, in spatial attention, after the input features F, two types of layers, $3 \times 1$ and $1 \times 3$ convolutional layers, are utilized to build two new feature maps $Q_y$ and $K_x$. Applying a softmax layer to the matrix multiplication of the transpose of the features captured in the vertical and horizontal directions yielded an intra-class spatial association as Eq. (4), where $\mathcal{S}_{(x,y)}$ denotes the $y$ position's effect on the $x$ position. The feature correlation matrix between any two points was computed and outputted by matrix multiplication; two similar spatial points increased one another, while two dissimilar spatial points decreased one another. This operation enables the network to optimally use and learn about the structure of various spatial locations. Subsequently, the softmax function was used on the correlation matrix in order to produce an attention map that represents the degree of similarity between each spatial position and the rest. Stronger similarity increases the response between two points. A channel attention map was created by adding a softmax layer between the input feature and transposing it on the channel-wise similarity map as Eq. (5). As a result, by conducting matrix multiplication, we obtained the channel dependence matrix. The channel dependency matrix is then applied to a softmax to improve the discrimination between the BAC structure and its background. These operations improved the expressiveness of class-dependent features by enhancing the contrast between them.

$$\mathcal{S}_{(x,y)} = \frac{\exp\left(Q_y^T \cdot K_x\right)}{\sum_{x'=1}^{N} \exp\left(Q_y^T \cdot K_{x'}\right)} \tag{4}$$

$$\mathcal{C}_{(x,y)} = \frac{\exp\left(F_x \cdot F_y^T\right)}{\sum_{x'=1}^{C} \exp\left(F_{x'} \cdot F_y^T\right)}, \tag{5}$$

## Loss function
One of the most difficult problems encountered was the high-class imbalance between BAC foreground pixels and background pixels. The loss in the BAC segmentation task was

computed from both the background and the BAC. Overall, the BAC pixels represented approximately 0.2% of the total pixels. To mitigate this issue, we experimented with various loss functions and optimizers. We tested several loss functions (MSE loss, cross entropy loss, dice loss), which showed good performance in other DL scenarios, but did not yield good results. Equation (6) shows the formula for the binary cross entropy (BCE) loss function. According to the equation, $T$ represents labels of a single image used as ground-truth, while $T_x$ represents a single element of $T$. The network's output prediction mask is shown by $P_x$. When using the BCE loss, the model tends to classify pixels into the unchanged type, which affects the result. To solve the data imbalance problem, we used BCEWithLogitsLoss, which worked well with imbalanced data because it gives each class the same amount of weight as if it had an equal portion of the dataset by the (pos weight) parameter (*Xiong et al., 2021*) Eq. (7). BCEWithLogitsLoss combines a sigmoid layer and the BCE loss. This approach was more numerically stable than using a plain sigmoid, followed by a BCE loss because it uses the log-sum-exp method for numerical stability by merging the operations into one layer. Adam was found to be the best optimizer for BCEWithLogitsLoss.

$$L_{BCE} = \sum_x -(T_x \log(P_x) + (1 - T_x) \log(1 - P_x)) \tag{6}$$

$$\ell(x, y) = L = \{l_1, \dots, l_N\}^\top, \\ l_n = -w_n[t_n \cdot \log \sigma(x_n) + (1 - t_n) \cdot \log(1 - \sigma(x_n))]. \tag{7}$$

## EXPERIMENTS

### Dataset

The University of Miami Institutional Review Board (IRB) approved this research study under ID number 20191227. The IRB also approved a waiver of consent and a HIPAA full waiver of authorization. Institutional guidelines were followed for the de-identification, extraction, and storage of 2,000 craniocaudal (CC) and mediolateral oblique (MLO) synthetic views of DBT exams performed in women aged 35 years and older, specifically over the time period from January 1, 2016, to December 31, 2018. The DBT exams were performed on six hologic dimensions mammography units located at three different breast imaging centers within the University of Miami Health System (UHealth), Miami, Florida. The DICOM data from UHealth were converted into 8-bit portable network graphics (PNG) format images. Following this, an expert radiologist specialized in breast imaging, who received fellowship training and had 10 years of experience, reviewed all the images and discarded images with poor quality due to artifact and those that were not CC or MLO synthetics views. The final dataset consisted of a total of 1,436 images. Most images were large-size with 2,457 pixels in height and 1,996 pixels in width. Each one was processed as a separate image. Figure 5 shows a flowchart of the dataset pre-processing steps.

The polygon tool of Supervisely (https://supervisely.com/) was used to manually draw BAC boundaries, as shown in Fig. 6. Following dedicated training, a medical student in their third year, a resident specializing in radiology in their second year, a breast imaging fellow, and an expert fellowship-trained breast imaging radiologist with 10 years of experience, each separately performed the BAC segmentation. After completing the

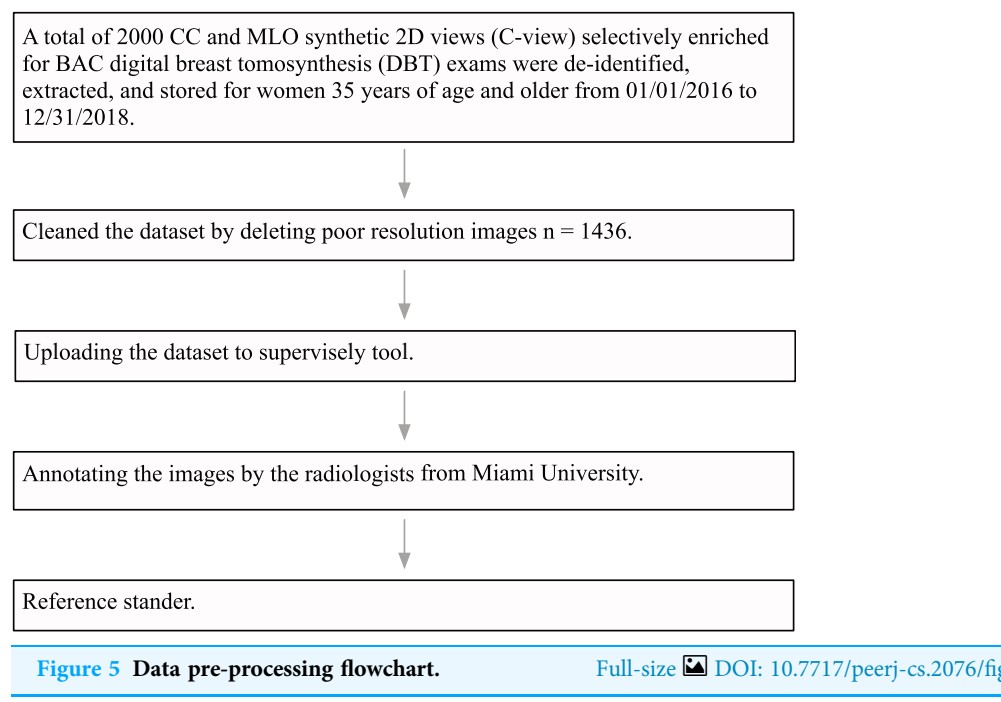

A total of 2000 CC and MLO synthetic 2D views (C-view) selectively enriched for BAC digital breast tomosynthesis (DBT) exams were de-identified, extracted, and stored for women 35 years of age and older from 01/01/2016 to 12/31/2018.

Cleaned the dataset by deleting poor resolution images n = 1436.

Uploading the dataset to supervisely tool.

Annotating the images by the radiologists from Miami University.

Reference stander.

**Figure 5  Data pre-processing flowchart.**     

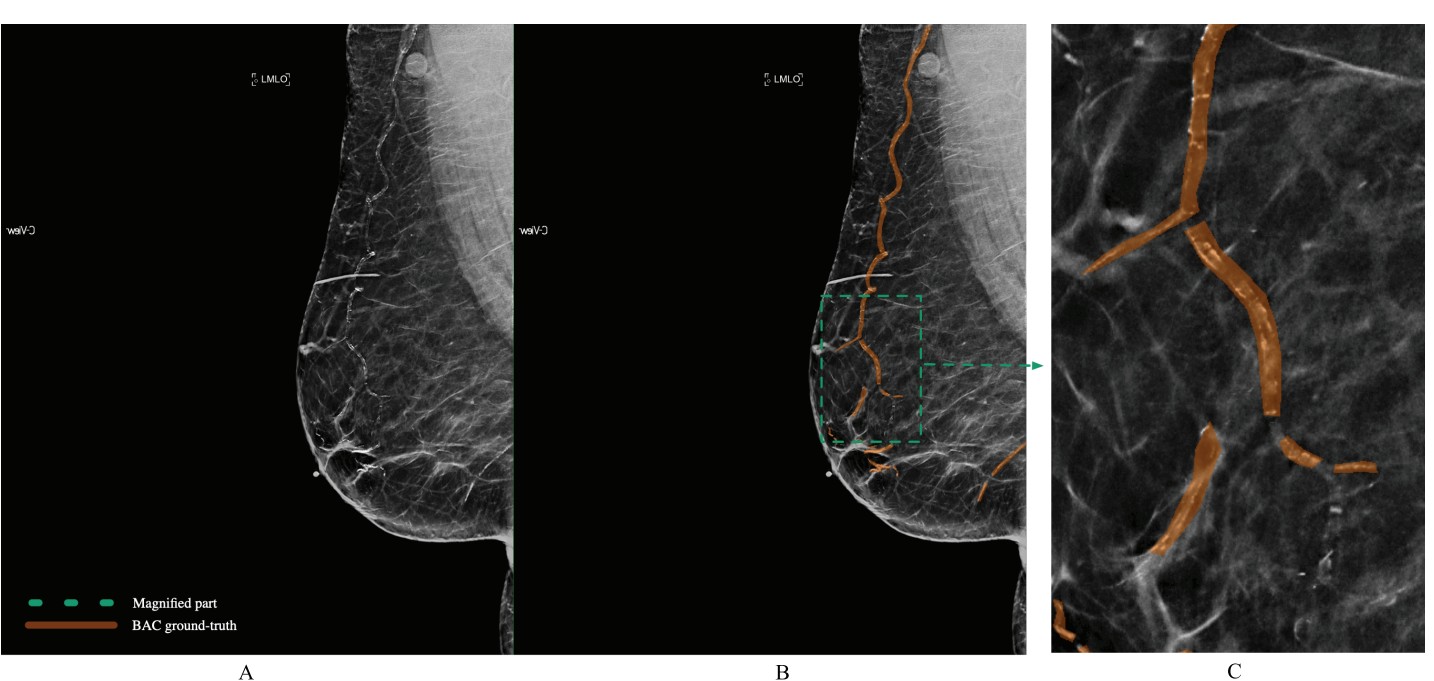

**Figure 6  Example of BAC ground-truth contoured by lines from a mammogram prepared by human experts.** The magnified part for the annotation is defined by the polygon. Mammogram and annotated image from our dataset.     

segmentations, the expert breast imaging radiologist evaluated each one and made any necessary modifications, to achieve gold-standard annotation. At this step, these BAC tracings were considered the ground-truth. After dividing the images into three separate

**Table 2 Summary of dataset statistics.**

| Dataset | Total number of images | Number of images with BAC | Number of images without BAC | Average BAC area to background area ratio | Overall average BAC area per image (%) |
|---|---|---|---|---|---|
| Train | 1,149 | 999 | 150 | 0.002406 | 0.209% |
| Validation | 143 | 122 | 21 | 0.002413 | 0.205% |
| Test | 144 | 135 | 9 | 0.002548 | 0.238% |

non-overlapping groups for our study 1,149 (80%) images for model training, 143 (10%) for validation, and 144 (10%) for testing, we meticulously analyzed various aspects of these datasets to gain deeper insights into the BAC class distribution and area. The findings from our analysis, including the total number of images, the distribution of images with and without the BAC, as well as the average BAC area to background area ratio, and the overall average BAC area per image, are comprehensively detailed in Table 2. The dataset and ground-truth have been made publicly available on GitHub (https://github.com/Manar-ibr/BacSeg/releases/tag/0.1.0).

## Evaluation criteria

The segmentation results were reported and evaluated through various widely used metrics, defined as follows:

$$\text{Accuracy} = \frac{TP + TN}{TP + FP + TN + FN} \tag{8}$$

$$\text{Precision} = \frac{TP}{TP + FP} \tag{9}$$

$$\text{F1-score} = \frac{2 \cdot \text{precision} \cdot \text{recall}}{\text{precision} + \text{recall}} \tag{10}$$

$$\text{Recall} = \frac{TP}{TP + FN} \tag{11}$$

$$\text{Jaccard index} = \frac{TP}{TP + FP + FN} \tag{12}$$

the terms TP, TN, FP, and FN represent the true positive, true negative, false positive, and false negative, respectively. The values were computed based on the pixel number. Nonetheless, when analyzing imbalanced datasets, accuracy can be deceiving, as it can be impacted by the classification results obtained for data that belongs to the majority class, which makes it more difficult for a classifier to perform well with data belonging to another class. For this reason, we used several metrics to deal with imbalanced datasets, including F1-score and Jaccard. Utilizing these metrics allows for a more comprehensive evaluation of a classifier's performance on imbalanced datasets. They can help identify models that not only perform well on the majority class but also maintain reasonable performance levels on the minority BAC class. The F1-score and Jaccard index are metrics used to calculate the ratio of TP pixels to the total number of detected pixels and were used to evaluate the overlapping between the predicted mask and ground-truth.

### Quantification metrics

To quantify BAC we used the metrics proposed by *Guo et al. (2021)*. The sum of the probability metric, known as PM, the sum of the mask area metric, known as AM, the sum of the mask intensity metric, known as SIM, the sum of the mask area with the intensity threshold X metric, known as T AMX, and the sum of the mask with the intensity threshold X metric, known as T SIMX. In the equations, m and n refer to the width and height of the mammogram, respectively, $P_{i,j}$ is the probability value at $<i, j>$ returned by the trained model, $I_{i,j}$ means the intensity value of the pixel at $<i, j>$, and $x$ is the intensity threshold.

$$AM = \sum_{i=0,j=0}^{m,n} 1_{pi.j>0.5} \tag{13}$$

$$PM = \sum_{i=0,j=0}^{m,n} P_{i,j} \tag{14}$$

$$SIM = \sum_{0 \leq i \leq m, 0 \leq j \leq n \,|\, p_{i,j>0.5}} I_{i,j} \tag{15}$$

$$T\,AMX = \sum_{0 \leq i \leq m, 0 \leq j \leq n \,|\, p_{i,j>0.5}} 1_{I_{i,j}>x} \tag{16}$$

$$T\,SIMX = \sum_{0 \leq i \leq m, 0 \leq j \leq n \,|\, p_{i,j>0.5}, I_{i,j}>x} I_{i,j}. \tag{17}$$

### Receiver operating characteristic curve

The ROC curve is a standard measure for comparing the ground-truth image and the output image of a segmentation method utilizing the confusion matrix. The variables involved in the generation of the confusion matrix are TP, FP, TN, and FN. To plot the ROC curve, sensitivity and 1_specificity are necessary measurements. The percentage of true positive pixels is the sensitivity, or true positive rate or recall. The 1_specificity or false positive rate, represents the ratio of false positive pixels. A higher percentage of sensitivity and 1_specificity ensures that the segmentation method is of high quality and performs well. The ROC plots the TP rate (sensitivity) with respect to the false positive rate. Only subregions that overlapped with the BAC ground-truth were considered TPs. All remaining subregions that did not overlap with the ground-truth BAC were classified as FP. The ROC curves were computed using sklearn.metrics.roc_curve().

## Experiment set-up

To demonstrate the performance of the RAU-Net model, we tested it on the mammogram dataset. For this implementation, Pytorch v1.12.1 frameworks were used. Because of the imbalance between the foreground and background, the binary cross entropy loss converges slowly. Based on the experiment results, we used BCEWithLogitsLoss. The models were developed using Google colab pro+, Python 3 Google Compute Engine backend (GPU) RAM: 51.01 GB Disk: 156.99 GB. The training and testing dataset images were reshaped into $640 \times 640$ to meet the requirements of the models. We enhance image contrast to emphasize the difference between the breast background tissue and the calcified

vessel. We normalize the input image to reduce the impact of variability in contrast between classified vessels and background, which improves model performance and makes it converge much faster. The dataset was normalized by subtracting the mean $\mu = 0.3$ and dividing by the standard deviation $\sigma = 0.2$. Our model was trained from scratch using the Adam optimization method, with $\beta1 = 0.5$ and $\beta2 = 0.999$. The batch size was set to 1, and the learning rate (lr) was initialized at 0.0001 and decayed after 20 epochs. Each model was trained with 70 epochs. At each epoch in the validation, we calculated the model score, which was the sum of the Jaccard index and dice coefficient; if the score was better than the previous score, then the weights were updated. Consequently, the best model based on the validation set was saved. The code of RAU-Net model has been made publicly available on GitHub (https://github.com/Manar-ibr/BacSeg.git).

## Ablation study

Because all previous methods' datasets were private, comparing them to an existing method is not feasible. Therefore, we defined three different comparisons to evaluate the performance of RAU-Net: the original U-Net, the U-Net with recurrent and residual to evaluate the effectiveness of the recurrent residual block, and the U-Net with Attention to evaluate the effectiveness of the attention module.

These models were utilized in the comparative process:

- **U-Net** (*Ronneberger, Fischer & Brox, 2015*)**:** the original model, which includes long-skip connections.
- **R2U-Net** (*Alom et al., 2018*)**:** the U-Net with recurrent residual blocks.
- **U-Net** (*Ronneberger, Fischer & Brox, 2015*) + **Attention block (AttU-Net):** the U-Net with attention block between the encoder and the decoder.

The same set-up as in our model was used for U-Net, R2U-Net, and AttU-Net.

## RESULTS

### Other analysis

The results derived from the 1,436 mammography images, as described in "Dataset", in terms of accuracy, sensitivity, F1-score, precision, and Jaccard values are presented as percentages in Table 3. As shown, the RAU-Net substantially outperformed the other models (as described in "Ablation Study"), with an overall accuracy of 99.8861%, a sensitivity of 69.6107%, a precision of 68.3948%, an F-1 score of 66.5758%, and Jaccard index of 59.5498%. The model achieved high overall accuracy, reflecting its effectiveness in correctly classifying the majority of pixels within mammograms. When compared to relevant models in the field, our model demonstrated promising results in terms of sensitivity, F-1 score, and Jaccard coefficient. These metrics, indicative of the model's ability to accurately identify true positives BAC and the precision and overlap between the predicted segmentation and ground truth, highlight the model's competitive performance despite the inherent challenges of the dataset. These challenges include imbalanced class distribution and the complex presentation of BAC. While the result achieved across these metrics emphasizes the model's good capabilities in BAC segmentation, there are areas for

**Table 3 The performance of the proposed RAU-Net is compared with relevant models on our annotated dataset.**

| Method | Accuracy | Sensitivity | Precision | F1-score | Jaccard index |
|---|---|---|---|---|---|
| U-Net (*Ronneberger, Fischer & Brox, 2015*) | 87.6021 | 21.0424 | 72.3476 | 30.8042 | 19.9893 |
| U-Net + Attention module (AttU-Net) | 90.4886 | 35.8634 | 40.6747 | 36.0990 | 24.1726 |
| R2U-Net (*Alom et al., 2018*) | 93.3955 | 40.0211 | 57.6745 | 39.8833 | 27.7960 |
| Our RAU-Net | 99.8861 | 69.6107 | 68.3948 | 66.5758 | 59.5498 |

**Note:**
The results are based on the test set.

**Table 4 Comparison of using various loss functions for RAU-Net.**

| Loss function | F1-score | Jaccard index | Sensitivity | Precision |
|---|---|---|---|---|
| Cross Entropy loss | 40.1226 | 29.4727 | 34.7386 | 61.1497 |
| MSE loss | 9.1688 | 5.1298 | 55.4237 | 5.3424 |
| Dice loss | 53.1895 | 40.3639 | 52.6647 | 62.2879 |
| BCEWithLogitsLoss | 66.5758 | 59.5498 | 69.6107 | 68.3948 |

improvement. Enhancing the sensitivity, for example, could be achieved by increasing the dataset size or employing more advanced data augmentation techniques. We compared the performance of the proposed RAU-Net using different loss functions (cross entropy loss, MSE loss, Dice loss, and BCEWithLogitsLoss). Table 4 shows that BCEWithLogitsLoss improved the segmentation results for F1-score, Jaccard index, precision, and sensitivity.

Figure 7 shows examples of BAC segmentation outputs using the RAU-Net model on mammography images. Examples are shown in Fig. 7 first column (A) and (D) where the RAU-Net correctly detects BAC comparable to human specialists. Figure 7 second column (B) and (E) displays a few mis-detected BAC results that the model incorrectly detects as BAC. Small clusters of non-continuous BAC can cause this. Figure 7 third column (C) and (F) illustrates how BAC was incorrectly detected as background in a few mammogram images. There is a possibility that this kind of error was made in the mammograms of individuals who had dense breast tissue. Dense breast tissue and the presence of clustered calcifications pose significant challenges for automated models like RAU-Net in distinguishing between normal anatomy and BAC. Dense breasts, characterized by a high proportion of fibrous and glandular tissue, can obscure mammographic details, making it difficult for automated systems to interpret the images accurately. If there are a sufficient number of situations like this for training, it is possible to prevent making this mistake. Figure 8 includes samples from the RAU-Net, U-Net, AttU-Net, and R2U-Net models results on our test dataset.

## Quantification metrics

The general evaluation metrics used in segmentation help evaluate the performance of the segmentation model by performing a pixel-to-pixel evaluation. This article aimed to quantify the amount of BAC within a mammogram into a single number that can be used to define BAC severity. To do this, we used the metrics by *Guo et al. (2021)*, and we
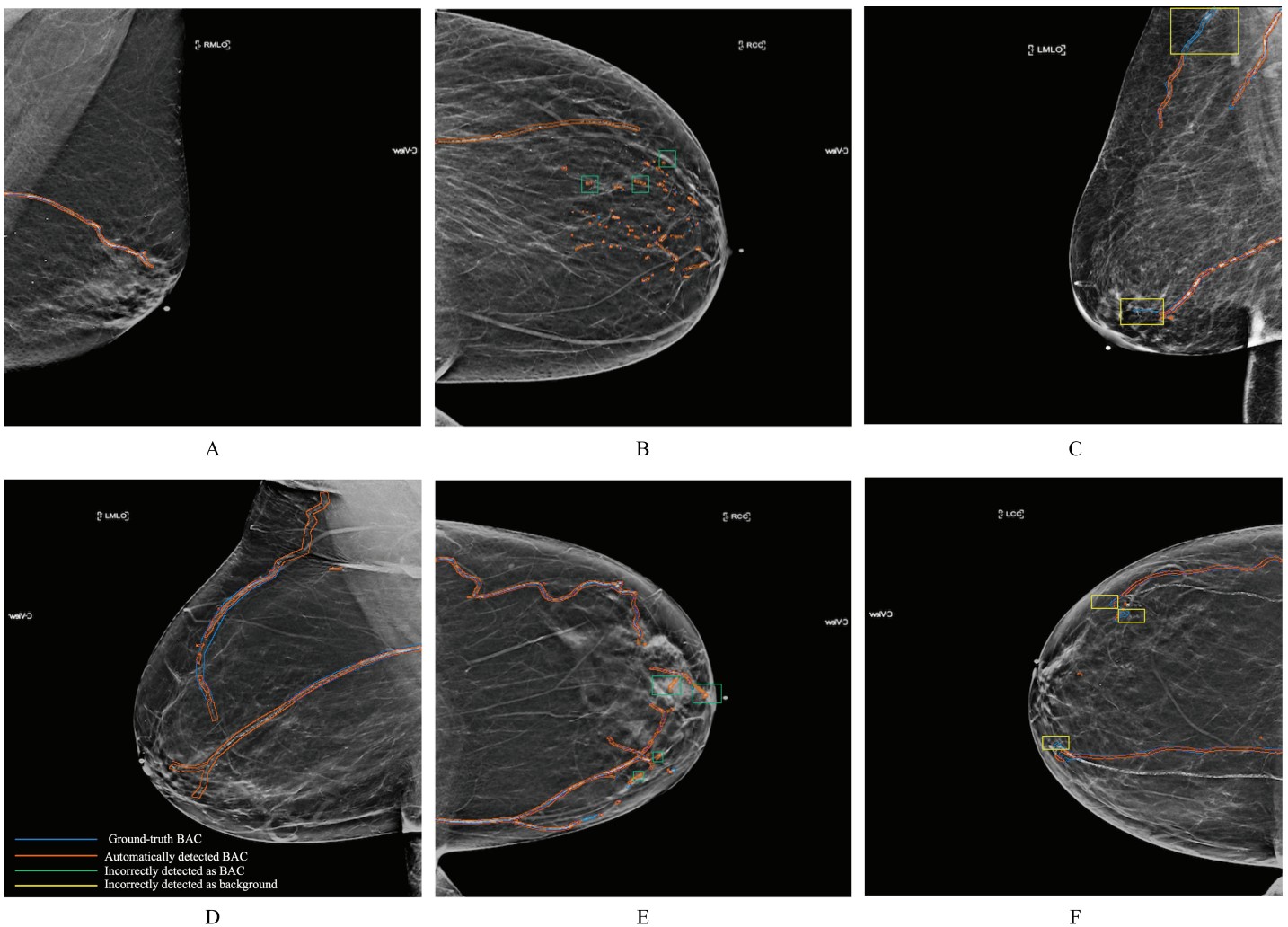

**Figure 7 Examples of synthesized 2D mammograms show RAU-Net's segmentation results.** Mammogram images from our dataset.

computed the correlation for five metrics using RAU-Net predicted mask compared to the same metrics that were produced based on the ground-truth mask. Table 5 presents a summary of the R2-correlation value for the five metrics. On the 144 test images, RAU-Net had the largest R2-correlation value of 0.83 between the predicted mask and ground-truth achieved when using the T AMX metric with threshold = 100, as in the scatterplot shown in Fig. 9. These findings coincide with the article's results (*Guo et al., 2021*). The evaluation of BAC quantification for the different stages of BAC is depicted in Table 6, in which the first stage includes no BAC, the second stage denotes minimal BAC, the third stage represents moderate BAC and the fourth stage for severe BAC. A higher quantification value is indicative of a more severe stage of BAC, indicating a greater risk of CVD.

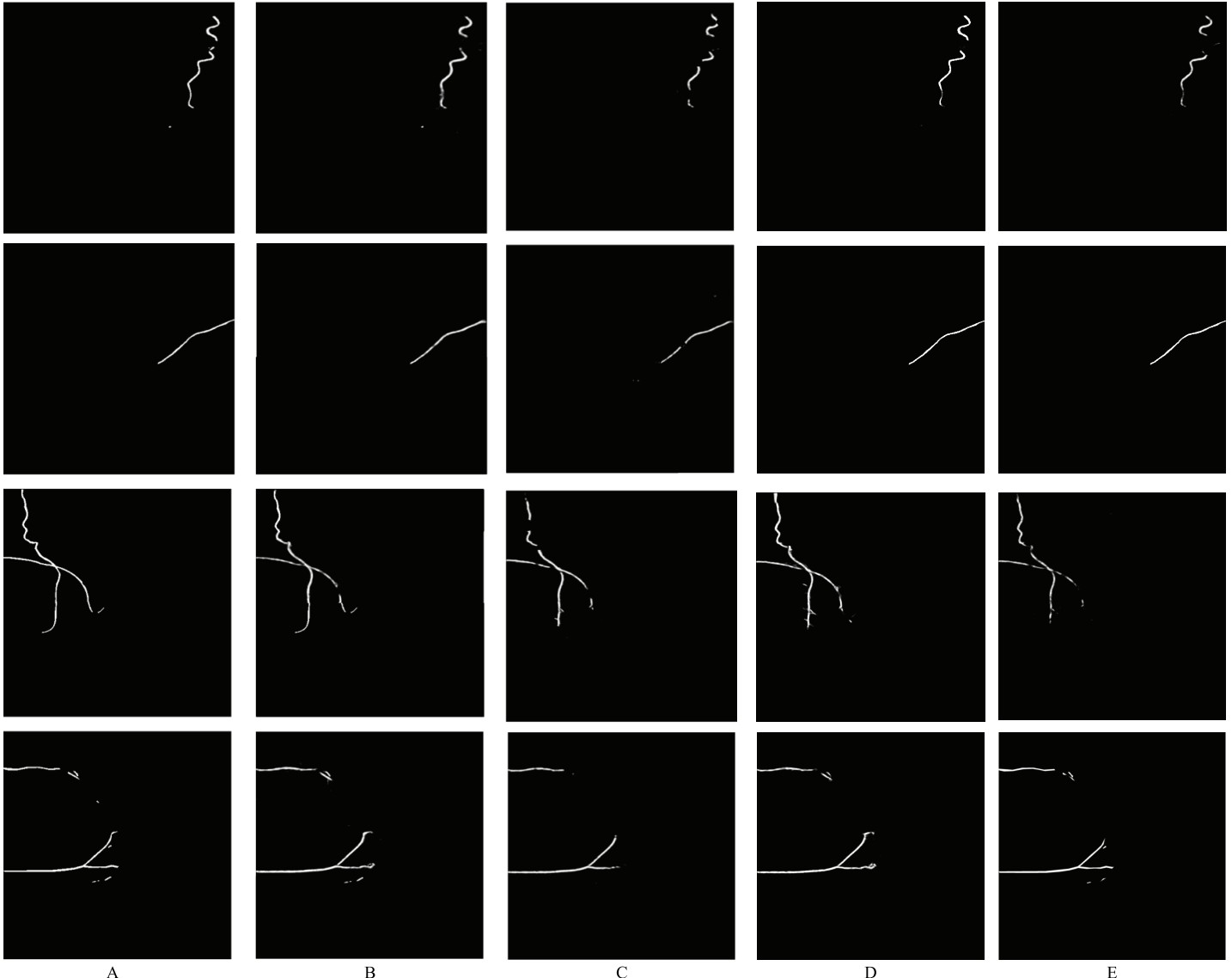

**Figure 8  Samples from the model results on our dataset.** (A) Shows the ground-truth image, (B) shows the mask generated using the RAU-Net model, (C) shows the mask generated using the U-Net model, (D) shows the mask generated using the AttU-Net model, and (E) shows the mask generated using the R2U-Net model. We generated these binary masks from mammogram images from our dataset.

## ROC analysis

Figure 10 shows the ROC curves of our proposed RAU-Net, U-Net, R2U-Net, and AttU-Net over our datasets for the segmentation of BAC. It can be comparatively illustrated *via* the local enlarged view that the suggested model outperformed all methods for the BAC segmentation task. The area under the curve (AUC) is a popular measure of classification accuracy. In general, higher AUC values indicate better performance. AUC can be interpreted as the average true positive rate over possible false positive rates. As shown, U-Net achieved AUC = 0.59. AttU-Net presented better BAC extraction ability compared to

**Table 5 Comparison of R2 correlation of the whole test set quantification metrics results for predicted masks and ground-truth.**

| Quantification method | R2-correlation |
| --- | --- |
| Sum of probability (PM) | 0.72 |
| Sum of mask area (AM) | 0.77 |
| Sum of mask intensity (SIM) | 0.82 |
| Sum of mask area with intensity threshold X (T AMX) | 0.83 |
| Sum of mask with intensity threshold X (T SIMX) | 0.71 |

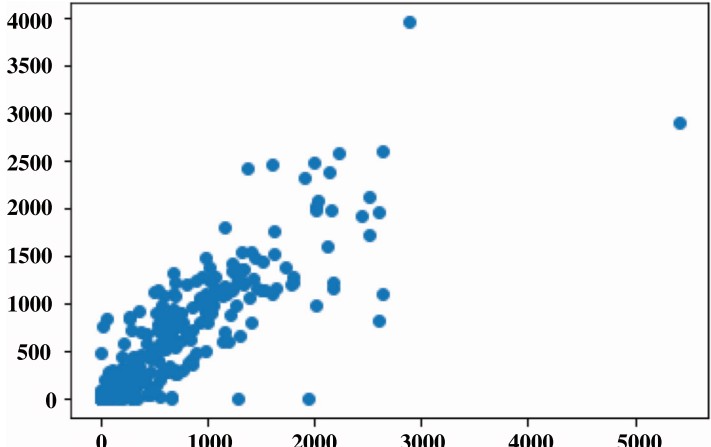

**Figure 9 Scatterplot of BAC area in pixels for all the data set, in which each point represents the BAC area for the image.** The x-axis represents the predicted values, while the y-axis represents the ground-truth values.

**Table 6 BAC with the five metrics in four patients.**

| | | | | |
| --- | --- | --- | --- | --- |
| **PM** | 11.756863 | 1,469.3569 | 2,823.2825 | 6,861.7925 |
| AM | 0 | 1,038 | 2,502 | 6,290 |
| SIM | 0 | 274,752 | 527,946 | 1,401,433.0 |
| TAM | 0 | 1,347 | 2,502 | 6,290 |
| TSIM | 0 | 274,752 | 527,946 | 1,401,433.0 |

**Note:**
From left: 0; no arterial calcification, 1; few arterial calcification, 2; moderate BAC, and 3; severe calcification affecting three or more vessels. The ground-truth BAC are contoured by lines, and the binary masks represent the predicted BAC masks. The severity scoring of BAC is based on *Hendriks et al. (2015)*, *McLenachan et al. (2019)*. We generated these binary masks from mammogram images from our dataset.

U-Net, particularly for small BAC, and achieved AUC = 0.75. The use of the attention mechanism on high-level features in both the channel and spatial dimensions has resulted in enhanced inter-class discrimination and intra-class aggregation. R2U-Net achieved AUC = 0.87 because of better feature representation for BAC segmentation tasks. When the values were compared, the best result, 0.96 AUC, was achieved by RAU-Net, which
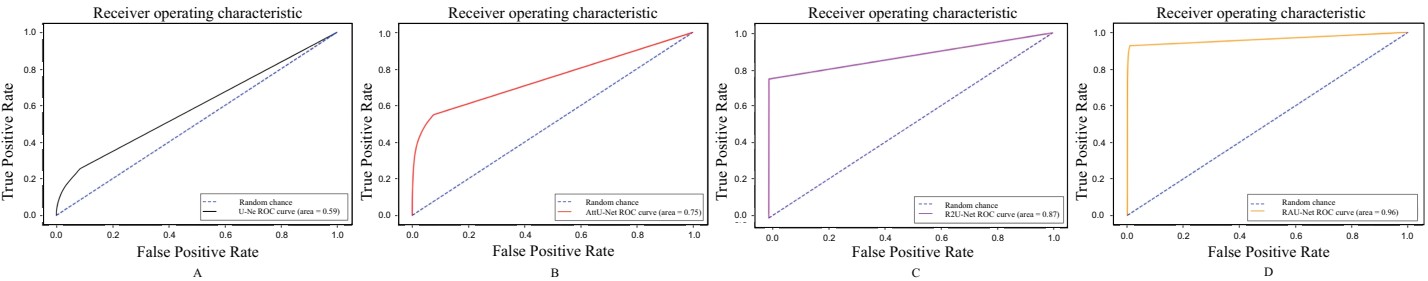

**Figure 10 ROC curves for segmentation BAC task (A) U-Net, (B) AttU-Net, (C) R2U-Net, and our proposed RAU-Net in (D).**

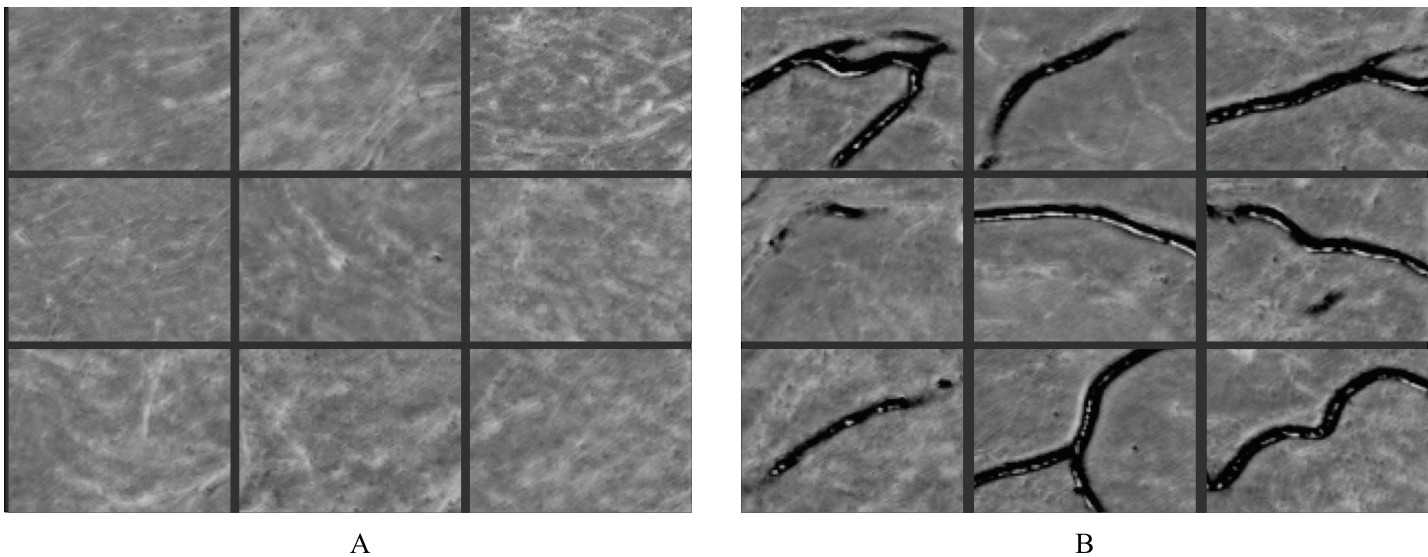

A

B

**Figure 11 Examples of learned features by the RAU-Net model from (A) non-BAC subset and (B) BAC subset.**

combined the advantages of recurrent blocks to enhance feature extraction and attention module to capture the features' long-range dependencies. This extension enhanced the result of detecting BAC, and it is a promising result in applying such a model clinically and using it for the estimation of CVD risk factors.

## Feature learning

The RAU-Net model is able to recognize features and make distinctions with high accuracy. Figure 11 illustrates some examples of feature maps that were created after the recurrent block's max-pooling layer in the contracting path. These layers were sufficiently deep to generate BAC classification features. Figure 11A demonstrates samples of learned features from non-BAC subset, whereas Fig. 11B shows samples of learned features from BAC subset. A comparison of both feature maps shows that the RAU-Net model learned the BAC samples at multiple scales and viewpoints. Moreover, the model recognized the

**Table 7 Comparison of segmentation time for the RAU-Net against other methods.**

| Method | Training time per epoch (S) | Testing time (S) |
|---|---|---|
| U-Net (*Ronneberger, Fischer & Brox, 2015*) | 313 | 132 |
| U-Net + Attention | 319 | 125 |
| R2U-Net (*Alom et al., 2018*) | 572 | 194 |
| Our RAU-Net | 589 | 191 |

absence of BAC in the non-BAC features and discriminated them from the other forms of calcifications seen in mammograms.

### Training and testing time

Table 7 shows a comparison between RAU-Net and the three related models (as described in "Ablation Study") in terms of training time per epoch and testing time for the test set. The fastest model was the U-Net, which was created as a segmentation model for medical images, but it does not capture the BAC after training for many epochs. AttU-Net required more time because of the computation of the attention module, which included matrix operations; in contrast, it enhanced the performance, as shown in Table 3. Adding recurrent blocks increased the computation time in the R2U-Net and RAU-Net, which was expected, given that it performed more computations in different time steps. Our RAU-Net model achieved better BAC segmentation results in a reasonable amount of time regarding both training and testing, 589 s to train one epoch and 191 s during the testing phase.

## DISCUSSION

Given the growing body of evidence regarding BAC as a risk marker for coronary artery disease, and its association with an increased risk of CVD events (*Iribarren et al., 2004*; *Ferreira, Szejnfeld & Faintuch, 2007*; *Maas et al., 2007*; *Iribarren & Molloi, 2013*; *Chadashvili et al., 2016*; *Yoon et al., 2019*; *Newallo et al., 2015*; *Margolies et al., 2016*), there is an interest in BAC segmentation and quantification to assist radiologists in reporting these calcifications in a standardized manner. However, BAC segmentation remains a technically arduous task. The literature is scarce regarding BAC segmentation using DL. The first published study for BAC segmentation in mammograms using DL was published by *Wang et al. (2017)*. A 12-layer CNN model was developed in a batch-based and pixel-wise manner. In another study, *AlGhamdi, Abdel-Mottaleb & Collado-Mesa (2020)* found that using U-Net shaped network improves model performance. They introduced DU-Net, which combines the short-dense and long-summation connections techniques.

In our study, we developed a new recurrent attention U-Net model, which embedded the recurrent block in the encoder and the decoder of the U-Net and used an attention module between the encoder and the decoder as mentioned in the previous sections. A major advantage of the recurrent block inside the model was that the accumulation of features over multiple time steps offered a more accurate and effective representation of features. Thus, it helped in detecting extremely low-level features that were required for the

BAC segmentation task. Furthermore, the attention module enhanced the network's ability to detect long-range dependency of breast arterial curvilinear structure and effectively used the multi-channel space for feature representation. The channel attention block enhanced the ability to distinguish between various features across different channels, thereby improving the model's discriminatory capabilities. The spatial attention mechanism allows the model to identify connections between features that are spatially distant but contextually related, ensuring that important patterns and relationships within the data are considered, regardless of their position. This ability for holistic data perception and identifying long-range dependencies enhances the model's efficacy, particularly in complex tasks like image segmentation, where understanding the overall structure and relation of parts within an image is essential.

Currently, there is no publicly available dataset for BAC, which necessitated our own generation of one consisting of de-identified mammograms known to have BAC. The ground-truth was created by experienced radiologists.

We compared our results with the classic U-Net model, the R2U-Net model, and the Attention+U-Net model. This comparison enabled us to assess the efficacy of each extension implemented on U-Net.

The RAU-Net performed better than the other models, according to the results reported in "Results". U-Net did not achieve good results because of BAC structures that were thin, long, and, occasionally, arborescent shaped. R2U-Net obtained better results than U-Net because of a better representation of low-level features. Adding attention to classic U-Net improved the performance, especially in terms of sensitivity. The best performance after our RAU-Net was R2U-Net, which learned good features. Our model used the recurrent block, which required the same number of network parameters as U-Net.

Another important task presented was the quantification of BAC. We used quantification measures to determine the level of calcification and found a strong correlation between the quantification values derived from the predicted mask and the ground-truth. Several different stages of the BAC were shown by calculating their quantification values. Further work is required to develop a good dataset divided into the different stages of BAC by a radiologist for further use to make a global BAC scoring.

Patients could be more adherent to treatment and health advice and make lifestyle changes if BAC on mammography is reported as evidence of coronary artery disease risk. That will prevent CVD at an earlier stage. A limitation of this work is that the model was trained on single mammogram views from synthesized 2D mammograms, which, as an imaging modality, is still not as widespread in use as DM. To address this limitation, we plan to evaluate our model on multiple mammogram views for each subject, with the intent to improve the accuracy of BAC quantification. We also plan to use the quantification values to generate a BAC score as minimal to mild, moderate, and severe.

## CONCLUSIONS

We developed a fully automated system for the segmentation and quantification of BAC in synthesized 2D mammograms to assist radiologists with reporting these calcifications in a

standardized manner. The model was evaluated using a set of images specifically collected for this task and annotated by medical students and radiology trainees trained and supervised by an expert breast imaging radiologist. The developed model was called RAU-Net. Our experiment confirmed that the recurrent and attention mechanism improved the BAC segmentation task by capturing long-range dependencies of the calcifications and using a multichannel space for effective feature representation. The proposed RAU-Net performed better for the BAC segmentation task when compared with the existing models, including the U-Net, R2U-Net, and attention-U-Net models. The quantitative metric demonstrated a strong correlation between predicted mask quantification values and the ground-truth values. Including the presence and quantification of BAC in mammography reports could add significant value to this type of imaging beyond breast cancer detection, and potentially help with CVD risk assessment and prevention in women, at no additional cost and no additional radiation exposure.

## PROMISING FUTURE WORKS

For future research, addressing the current limitations and advancing the study of BAC quantification using mammograms requires a multi-faceted approach. Incorporating 3D imaging technologies like DBT could significantly improve the depth and accuracy of BAC analysis. A longitudinal approach to tracking BAC progression over time would provide invaluable insights into the studies of cardiovascular risks. Developing an AI-driven tool that leverages quantified BAC scores for personalized cardiovascular disease risk prediction could transform patient care by enabling early and tailored interventions. Implementing advanced machine learning strategies, including multi-view learning, is crucial for enhancing the model's accuracy and generalizability. The integration of BAC scoring systems into clinical decision support frameworks could support healthcare professionals in making more informed decisions, potentially leading to improved management of cardiovascular risks. Through these comprehensive research directions, future studies aim to refine diagnostic capabilities, personalize risk assessments, and ultimately improve outcomes for individuals at risk of cardiovascular diseases associated with BAC.

## ACKNOWLEDGEMENTS

The authors would like to thank Laura C. Figueroa-Diaz, Taylor A. Schwartz, and Tiffany Eatz from the University of Miami Miller School of Medicine, Miami, FL, USA, for their contribution in annotating the dataset. We also thank Tiffany Eatz for her help with language and style editing.

### Funding

This research work was funded by the Makkah Digital Gate Initiative under Grant No. (MDP-IRI-6-2022) and received technical and financial support from Emirate of Makkah Province and King Abdulaziz University, DSR, Jeddah, Saudi Arabia. The funders had no

role in study design, data collection and analysis, decision to publish, or preparation of the manuscript.

## Grant Disclosures
The following grant information was disclosed by the authors:
Makkah Digital Gate Initiative: MDP-IRI-6-2022.
Emirate of Makkah Province and King Abdulaziz University, DSR, Jeddah, Saudi Arabia.

## Competing Interests
The authors declare that they have no competing interests.

## Author Contributions
- Manar AlJabri conceived and designed the experiments, performed the experiments, analyzed the data, performed the computation work, prepared figures and/or tables, and approved the final draft.
- Manal Alghamdi conceived and designed the experiments, analyzed the data, authored or reviewed drafts of the article, and approved the final draft.
- Fernando Collado-Mesa conceived and designed the experiments, analyzed the data, authored or reviewed drafts of the article, and approved the final draft.
- Mohamed Abdel-Mottaleb conceived and designed the experiments, analyzed the data, authored or reviewed drafts of the article, and approved the final draft.

## Ethics
The following information was supplied relating to ethical approvals (*i.e.*, approving body and any reference numbers):

The University of Miami Institutional Review Board (IRB) approved this research study under ID number 20191227. The IRB also approved a waiver of consent and a HIPAA full waiver of authorization.

## Data Availability
The code is available at Zenodo: Aljabri, M. (2024). Manar-ibr/BacSeg: Initial release. Zenodo. https://doi.org/10.5281/zenodo.10515261.

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
