# Peer review of "Recurrent attention U-Net for segmentation and quantification of breast arterial calcifications on synthesized 2D mammograms"

_PeerJ Computer Science, doi:10.7717/peerj-cs.2076_

## Round 0.1 · original submission · Major Revisions

The reviewers recommended major revision to your article. You are are required to address all the comments and suggestions and resubmit a revision.

**Language Note:** The review process has identified that the English language must be improved. PeerJ can provide language editing services - please contact us at [email protected] for pricing (be sure to provide your manuscript number and title). Alternatively, you should make your own arrangements to improve the language quality and provide details in your response letter. – PeerJ Staff

Reviewer 1 ·

Basic reporting

- This paper presents a method that could help radiologists in detecting and quantifying BAC in synthesized 2D mammograms using deep learning techniques.

- This paper is well-written, but it needs several improvements.

- Create sub-section in Introduction (section-1) to describe "Contributions" in one or two paragraphs.

- Separate related works completely as one section (i.e., section-2).

- Why there is some ambiguity in section numbering (Experiment section is showing as 1) ?

- Provide a GitHub link for newly created dataset.

- Include a figure to highlight an end-to-end proposed approach.

Experimental design

- Include "Ablation study" as a separate sub-section in Experimental section.

- Create a table to show the final statistic of the dataset (both before and after pre-processing).

- Compare your approach with recent methods and state-of-the-art works (i.e., sub-section: 1.4 Comparison)

Validity of the findings

- Include precision and recall in Table-3.

- Add promising future works after Conclusion section.

- Upload code on Github and provide the link in the revised manuscript.

Additional comments

Major revision is required.

Reviewer 2 ·

Basic reporting

In abstract, it would benefit from additional clarity regarding the dataset used, including its source and any preprocessing steps applied.

Regarding the recurrent attention U-Net model presented in the manuscript , it would be helpful to provide more insights into the interpretability of the attention mechanism.
Discussing how the attention module contributes to feature selection and classification could enhance the understanding of the model's performance and applicability.

The evaluation metrics report overall good accuracy but relatively lower sensitivity, F-1 score, and Jaccard coefficient.
It would be valuable to delve deeper into the reasons behind these performance discrepancies and discuss potential strategies for improving sensitivity and overall segmentation quality.

Experimental design

Authors need to provide a more detailed discussion on how the recurrent attention U-Net outperforms or complements existing approaches would strengthen the manuscript's contribution to the field of breast arterial calcification detection and quantification.

Include information about any potential biases or limitations associated with the dataset, such as demographic characteristics or imaging artifacts, to provide context for the model's performance evaluation.

The involvement of multiple annotators in BAC segmentation and subsequent evaluation by an expert radiologist enhances the reliability of the ground-truth annotations. However, it would be helpful to discuss any inter-annotator variability and steps taken to ensure consistency among annotators, as this could impact the robustness of the segmentation model.

The evaluation criteria section provides a clear explanation of the metrics used to assess the segmentation performance. However, it would be beneficial to include a brief rationale for selecting these specific metrics and how they address the objectives of the study, especially given the challenges posed by imbalanced datasets.

Validity of the findings

It would be beneficial to discuss any sensitivity analyses or hyperparameter tuning conducted to optimize the model's performance, as this information could inform future implementations or extensions of the proposed approach.

Authors may discuss the clinical significance of the achieved segmentation and quantification results, particularly in relation to the diagnosis and management of cardiovascular disease risk factors associated with breast arterial calcifications (BAC).

The visualization of BAC segmentation outputs in Figure 6 provides valuable insights into the model's performance. However, it would be beneficial to include additional qualitative analyses or discussions on specific challenges or limitations encountered by the model in accurately detecting BAC, particularly in cases of small or non-continuous BAC clusters.

The analysis of feature learning in the RAU-Net model, illustrated in Figure 10, provides valuable insights into the model's ability to recognize distinguishing features of BAC. Authors are required to discuss how these learned features contribute to the model's overall segmentation and quantification performance, particularly in terms of distinguishing BAC from other types of calcifications or anatomical structures.

Additional comments

Proofread the text thoroughly to catch any grammatical errors, typos, or inconsistencies. Consider using grammar-checking tools or seeking feedback from colleagues to ensure the highest quality of writing.

Some sentences are quite lengthy and complex, which might make them difficult to follow. Consider breaking them down into smaller, more digestible sentences for better readability.

---

## Round 0.2 · accepted · Accept

I am pleased to inform you that your paper has been accepted for publication in PeerJ Computer Science. Your manuscript has undergone two-rounds of rigorous peer review, and I am delighted to say that it has been met with high praise from our reviewers and editorial team. Your research makes a significant contribution to the field, and we believe it will be of great interest to our readership. On behalf of the editorial board, I extend our warmest congratulations to you.

Reviewer 1 ·

Basic reporting

Authors have addressed all comments in the revised manuscript.

Experimental design

Authors have addressed all experiment related comments like inclusion of evaluation metrics (Precision and Recall), ablation study, etc., in the revised manuscript.

Validity of the findings

Authors have addressed all comments in the revised manuscript. It can be accepted in the present form.

Additional comments

Accept

Reviewer 2 ·

Basic reporting

All the comments have been adressed. May accept as it.

Experimental design

no comment

Validity of the findings

no comment

Additional comments

no comment